# Architectural Experiment Design of Solar Energy Harvesting: A Kinetic Façade System for Educational Facilities

## Ho Soon Choi

Department of Architecture, Gachon University, 1342 Seongnamdaero, Sujeong-gu, Seongnam-si 13120, Korea; hosoon@gachon.ac.kr; Tel.: +82-31-750-5519

**Abstract:** This study proposes an architectural design for renewable energy production to increase energy independence in the architectural field. Among natural energy sources, solar panels that can be applied to building façades have been developed to use solar energy. To maximize renewable energy generation, solar panels can be adjusted according to the optimal tilt for each month. They can be attached to and detached from the building façade and installed on an existing building elevation. Thus, it is possible to increase the energy independence of old buildings. The solar panel developed in this study increases energy independence and presents a creative "kinetic façade," in which solar panels move each month according to the optimal tilt angle.

**Keywords:** architectural experiment design; solar energy harvesting; solar panel; kinetic façade system; renewable energy





## 1. Introduction

In response to increasingly severe global pollution, environmental protection solutions are being sought worldwide. The United Nations announced the Sustainable Development Goals (SDGs) policy and set a goal of "enduring access to affordable, reliable, sustainable, and modern energy for all" by 2030 [1]. Air pollution is the most urgent problem to be addressed because it has immediate health effects. Policies to reduce greenhouse gas (GHG) emissions by reducing the use of fossil energy and to increase renewable energy production are actively being promoted at the global level [2,3].

In the building industry, building and building-construction sectors are responsible for almost one-third of the total global energy consumption and nearly 15% of direct $CO_2$ emissions. Energy demand from buildings and building construction continues to increase, driven by improved access to energy in developing countries, growing demand for air conditioning in tropical countries, greater ownership and use of energy-consuming appliances, and rapid growth in global building floor area [4]. As the building sector greatly impacts the global environment, reducing building energy use by converting to renewable energy can reduce global environmental pollution. From this perspective, an experimental architectural design to increase the energy independence of buildings was developed in this study through generation of renewable energy from natural sources.

The technology to produce renewable energy from natural energy is called energy harvesting. It uses kinetic energy, such as micro vibration, and natural energy sources, such as solar, geothermal, and wind energies, to generate renewable energy. Recently, the scope of energy harvesting research is expanding, such as the development of storage devices for renewable energy generated with energy harvesting and material research to increase energy generation efficiency [5–7].

This study proposes an energy harvesting architectural design that uses solar energy sources, and the renewable energy generated through architectural design is supplied to the building to increase its energy independence. Eco-friendly architectural designs based on an energy harvesting principle can produce optimal energy with application to a building façade and help create a new urban landscape by contributing to building aesthetics.

## 2. Research Scope and Subject

### 2.1. Educational Facilities in South Korea

This study examines elementary schools in South Korea to develop an eco-friendly architectural design for renewable energy production. Elementary schools were chosen as the research subject because there are many more elementary schools in South Korea than other educational facilities, and the number is increasing (Table 1). Among the educational facilities that have been in existence for over 41 years, elementary schools account for 15.2%, middle schools account for 13.7%, and high schools account for 14.2% (Figure 1).

**Table 1.** Number of elementary, middle, and high schools in South Korea by year [8].

| Category | 2017 | 2018 | 2019 | 2020 | 2021 |
|---|---|---|---|---|---|
| Elementary school | 6040 | 6064 | 6087 | 6120 | 6157 |
| Middle school | 3213 | 3214 | 3214 | 3223 | 3245 |
| High school | 2360 | 2358 | 2356 | 2367 | 2375 |

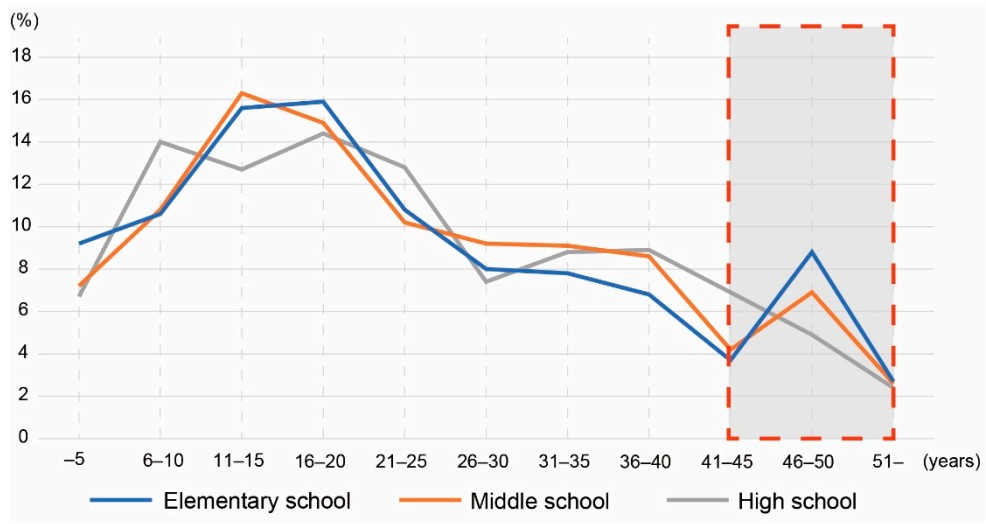

**Figure 1.** Status percentage of age of educational facilities as of 2018 in South Korea [9].

As elementary school buildings are the oldest, remodeling is urgently needed for safety and energy efficiency. Elementary schools can easily use developed architectural designs; large-scale renewable energy can be secured through eco-friendly designs applied to elementary school buildings. The architectural characteristics of elementary schools were also significant in selection for this study. Elementary schools in South Korea were standardized and designed according to the "standard blueprint" issued by the Ministry of Education in 1962. Although the standard blueprint system was abolished in 1997, basic designs including building and playground layouts are still based on standard blueprints [10]. Consequently, in terms of building size, the total floor area is determined according to the number of students based on the required building area per student with reference to the standard blueprints. As of 2022, the floor area per elementary school student in South Korea is 17.1 m$^{-2}$ [11].

Regarding the layout and shape of the building, the main elementary school building generally faces south and has a large playground in front of the building (Figure 2 left). As there is no building in the front area; solar energy-based architectural designs with sufficient sunlight can be used in elementary school design. The classrooms in the building use modules with a uniform structure. Thus, the classroom width is projected on the outside of the building, which has a standardized appearance (Figure 2 right).

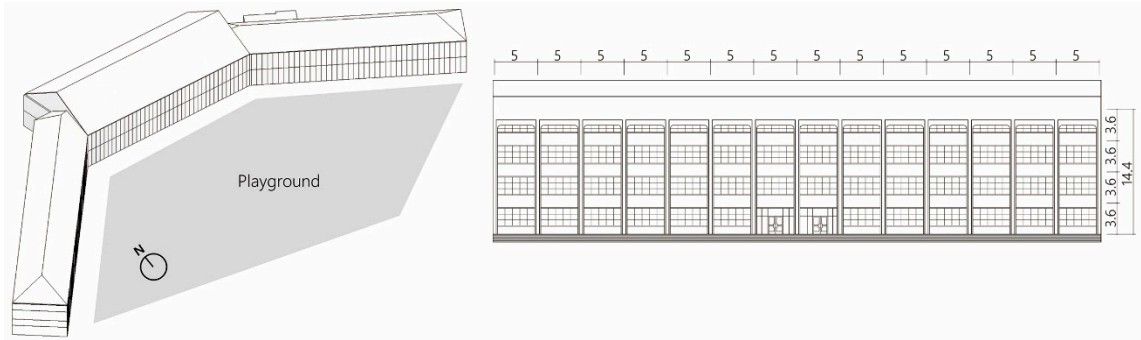

**Figure 2.** Layout and standardized elevations of elementary schools in South Korea.

### 2.2. Research Subjects

Won-Hyo Elementary School (37.5° N, 126.9° E) in Seoul and Cho-Rang Elementary school (35.1° N, 129.0° E) in Busan were selected as the study targets (Figure 3). Won-Hyo Elementary School, built in 1983, consists of a main building and sports facility. The main building is five stories with a total floor area of 7589 m$^{-2}$, and the front of the building faces southeast. As of 2022, there are a total of 622 building users, 572 elementary school students, and 50 teachers [12]. Cho-Rang Elementary school, built in 1937, consists of a main building of four stories. The front of the building faces southeast. Total floor area is 8218 m$^{-2}$. As of 2022, there are a total of 318 building users, 292 elementary school students, and 26 teachers [13].

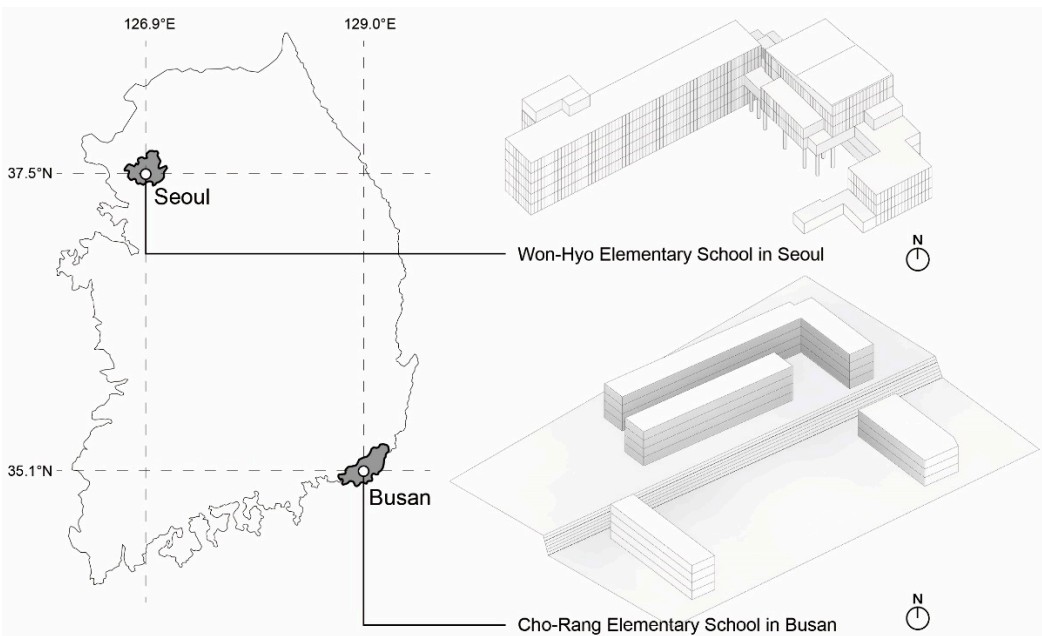

**Figure 3.** Research subjects: Won-Hyo Elementary School in Seoul and Cho-Rang Elementary School in Busan, South Korea.

Won-Hyo and Cho-Rang Elementary Schools are more than 40 years old and need remodeling. There is a playground in front of the main building, which is a typical elementary school layout in South Korea. Consequently, the building façade receives sufficient light.

## 3. Kinetic Façade Design and Operating Concept

### 3.1. Solar Energy Harvesting

Energy harvesting enables self-powered electronic devices to collect ambient energy from natural energy sources [14]. Solar energy-based energy harvesting is a representative method for producing renewable energy using solar panels [15,16].

Research on solar panels has focused mainly on materials that increase energy production efficiency; optimization of solar cells on solar panels and energy harvesting are representative examples [17–19]. Solar energy harvesting technology is actively used in the construction field [20,21]. A study has been conducted on solar thermal panels that generate hot water using solar heat as a material for exterior building walls, with photovoltaic (PV) technology applied to produce renewable energy from sunlight [22,23]. PV used for exterior walls of buildings decreases energy generation efficiency when its surface temperature increases owing to sunlight. To increase energy generation efficiency, research has been conducted to develop a cooling device to lower the temperature of PV modules [24,25]. Research on PV materials has resulted in eco-friendly exterior building materials. PV material is transparent, and integrated with building glass; it was developed to produce renewable energy and act as a shading screen [26–28]. PV material can be used as a main architectural design element. Solar panels are usually installed on the roofs of buildings because they are easily installed and maximize exposure to sunlight [29]. Solar panels installed on rooftops cannot be used as architectural design elements.

This study considers application of solar energy harvesting technology to architectural design. A new architectural design is proposed, with detachable solar panels applied to an existing building façade. To maximize renewable energy production, the solar panels are adjustable to the optimal angle to the sun. As the solar panels are installed on the building façade, roof installation is excluded from this study.

### 3.2. Study of Solar Radiation for Installation of Solar Panels

A solar radiation survey was conducted on the façade of the building, excluding the roof. To maximize the energy generation efficiency, solar panels should be installed at the optimal location on the building elevation. This study used the building information modeling (BIM) technology to create a building in a virtual space and measure the energy production of installed solar panels. BIM is used in the field of architecture to design, informatize, and visualize virtual buildings in three dimensions [30,31]. Digital technology for realizing virtual buildings was first developed in 1974 and was known as BDS (Building Description System) [32]. BIM technology has been used to simulate buildings in a virtual space since 1992 [33].

Autodesk Revit 2020 software, a widely used BIM program, was used in this study to create Won-Hyo and Cho-Rang Elementary School in a virtual space [34]. Revit creates a building in 3D, including all building information. Insight, a Revit software plug-in, is a program that simulates solar irradiation and energy generation for a virtual building created in Revit [35].

Figure 4 shows the simulation results for global radiation of the building façade at the Won-Hyo Elementary School in Seoul (37.5° N, 126.9° E) and Cho-Rang Elementary School in Busan (35.1° N, 129.0° E) during optimal bright sunlight hours from 1 January to 31 December 2021. The simulation results indicate that the building elevations receiving optimal solar radiation face south and southeast.

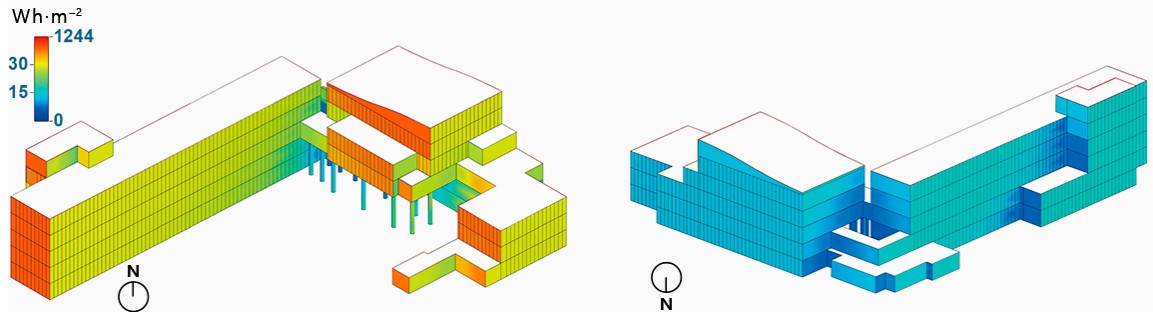

**Figure 4.** Global solar radiation on all façades of Won-Hyo Elementary School in Seoul (upper image) and Cho-Rang Elementary School (lower image).

### 3.3. Optimal Tilt Angle (βo) of Solar Panel for Maximal Energy Generation

In modern construction, solar panels are installed to obtain renewable energy from the sun, usually at a fixed angle on the roof or façade of a building. For optimal energy generation efficiency, the installation angle of the solar panel is adjusted according to the location of the sun. A fixed solar panel cannot optimally generate energy. The optimal tilt angle (βo) of the solar panel was calculated considering the latitude and solar declination at Won-Hyo and Cho-Rang Elementary School (Equation (1)) [36].

$$\beta o = a_1 + a_2 \varphi,\tag{1}$$

where *βo* represents the optimal tilt angle; $\varphi$ is the latitude (37.5° N) of Won-Hyo Elementary School and (35.1° N) of Cho-Rang Elementary School; and $a_1$ and $a_2$ are coefficients derived from the solar declination, as shown in Table 2. The energy generation efficiency of a solar panel is maximized when it is situated at the optimal tilt angle (βo).

**Table 2.** Coefficients ($a_1$, $a_2$) based on solar declination and optimal tilt angle (βo).

| Month | January | February | March | April | May | June | July | August | September | October | November | December |
|---|---|---|---|---|---|---|---|---|---|---|---|---|
| Solar declination (deg) | −21.269° | −13.289° | −2.819° | 9.415° | 18.792° | 23.314° | 21.517° | 13.784° | 2.217° | −9.599° | −19.148° | −23.335° |
| $a_1$ | 31.33 | 16.25 | 6.80 | −6.07 | −14.95 | −19.27 | −15.65 | −4.23 | 6.42 | 15.84 | 23.61 | 30.56 |
| $a_2$ | 0.68 | 0.86 | 0.84 | 0.87 | 0.87 | 0.87 | 0.83 | 0.75 | 0.77 | 0.83 | 0.84 | 0.76 |
| Seoul. βo (deg) | 57° | 49° | 38° | 27° | 18° | 13° | 16° | 24° | 35° | 47° | 55° | 59° |
| Busan. βo (deg) | 55° | 47° | 35° | 24° | 16° | 11° | 14° | 22° | 34° | 45° | 53° | 57° |

### 3.4. Solar Panel Design According to Optomal Tilt Angle (βo)

For maximum energy generation efficiency, the optimal tilt angles from January to December were calculated for installed solar panels at the study site. Using the Revit 2020 software, solar panels were installed on the building façade according to the optimal tilt angle. The solar energy generated in 1 year was calculated using Insight, a plug-in program in Revit [37]. Figure 5 shows an example of solar energy generation (2691 kWh) for 1 year with a solar panel (2 m × 1 m) installed on the building façade of the Won-Hyo Elementary School.

The solar panel module can be oriented in two ways to fit the elementary school (2 m × 1 m) building façade.

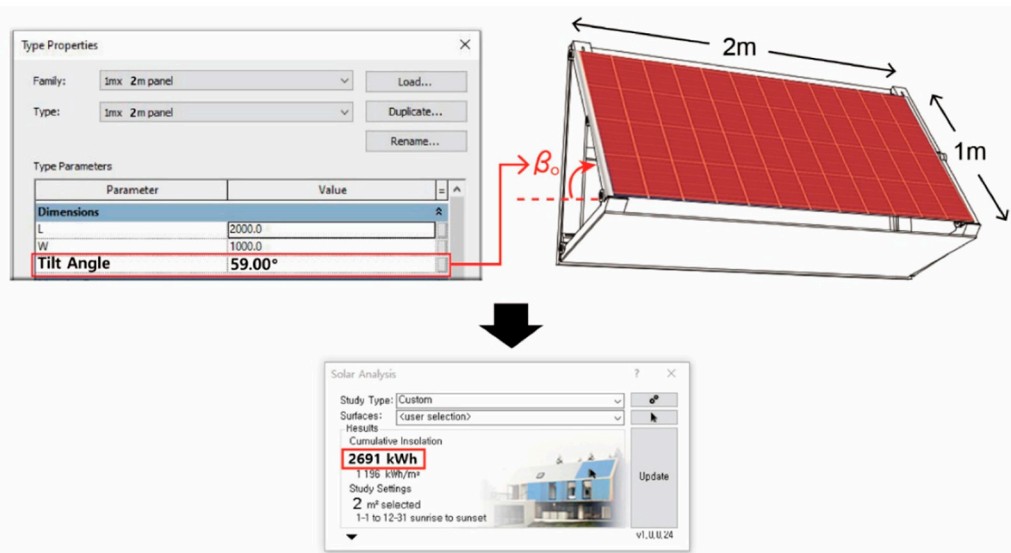

**Figure 5.** Example of solar energy generation (2691 kWh) of solar panel according to optimal tilt angle (59°) using Revit 2020 and Insight software.

Type A corresponds to 2 m in width and 1 m in length; Type 2 corresponds to 1 m in width and 2 m in length (Figure 6).

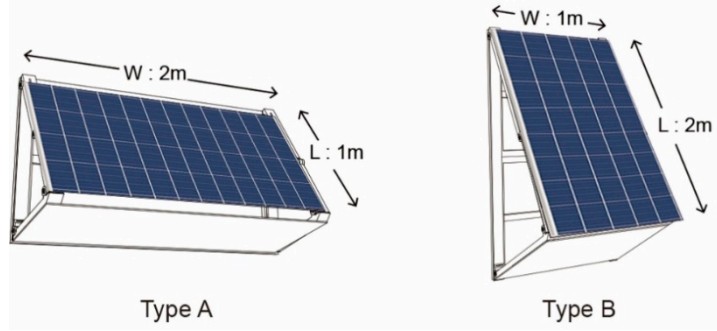

**Figure 6.** Two types of solar panel.

The Hanwha Q. PEAK DUO ML-G11.5/BFG solar module was selected as the solar panel (Table 3) [38]. This model can be used for both residential and commercial buildings; the model size (2 m × 1 m) fits the Won-Hyo and Cho-Rang Elementary School building façade and can be easily installed.

**Table 3.** Hanwha Q.PEAK DUO ML-G11.5/BFG solar module.

| Type | Parameter | Value |
|---|---|---|
| | Model | Q.PEAK DUO ML-G11.5/BFG |
| | Length (mm) | 2000 |
| Solar module | Width (mm) | 1000 |
| | Power capacity (kW/unit) | 500 Wp |
| | Efficiency (%) | 21% |

*3.5. Kinetic Façade System Design and Solar Energy Generation*

In a kinetic façade system, for optimal solar energy generation, a solar panel is placed at the top and an aluminum panel with high reflectivity at the bottom. The upper solar panel is adjusted every month to the optimal tilt angle. With high reflectivity, the lower aluminum panel reflects indirect light to the solar panel (Figure 7).

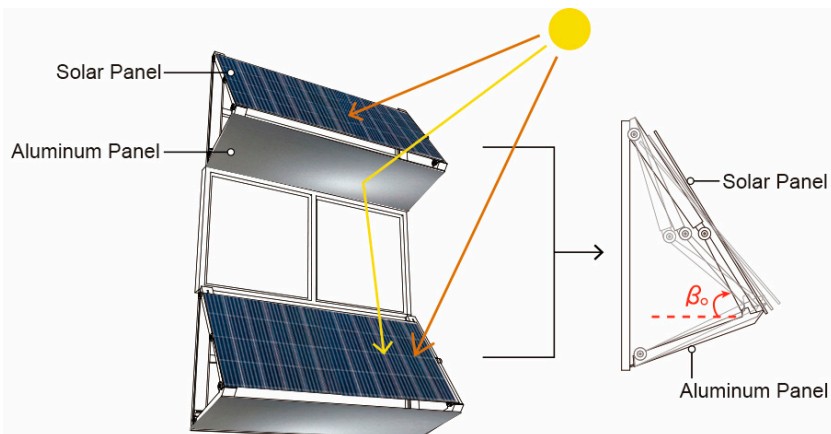

**Figure 7.** Principle of kinetic façade system with solar panel according to optimal tilt angle ($\beta o$).

To determine the shape of the solar panel to be placed at the top of the kinetic façade system, Type A and Type B solar panels (Figure 6) were fitted to be suitable for the basic module (2 m $\times$ 1 m) of the Won-Hyo and Cho-Rang Elementary School building façades. Using the Revit software, Type A (2 m $\times$ 1 m) and Type B (1 m $\times$ 2 m) kinetic façade systems were created on the south- and southeast-facing building façades, which receive optimal solar radiation. The amount of solar energy generated from January to December was calculated for the two types of kinetic façade systems using the Revit plug-in program Insight.

Figure 8 shows a three-dimensional image of Won-Hyo and Cho-Rang Elementary school with Type A and Type B kinetic façade systems.

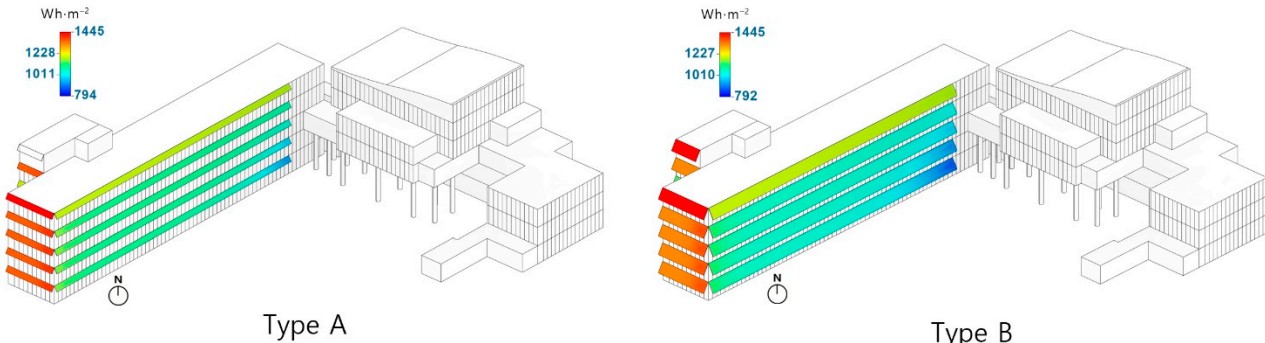

**Figure 8.** Simulation of solar energy generation in December 2021 with Type A and Type B kinetic façade systems installed on south- and southeast-facing building façades of Won-Hyo Elementary School in Seoul (upper image) and Cho-Rang Elementary School in Busan (lower image).

In the case of Won-Hyo Elementary School, Type A solar panels placed on the south and southeast façades resulted in a total solar area of 420 m$^{-2}$. For Type B, the total solar area was 800 m$^{-2}$. The total solar energy generation from January to December 2021 was 5,663,989 and 10,143,682 kWh with Types A (420 m$^{-2}$) and B (800 m$^{-2}$) solar panels, respectively.

In the case of Cho-Rang Elementary School, Type A solar panels placed on the south and southeast façades resulted in a total solar area of 820 m$^{-2}$. For Type B, the total solar area was 1640 m$^{-2}$. The total solar energy generation from January to December 2021 was 11,947,833 and 21,900,856 kWh with Types A (820 m$^{-2}$) and B (1640 m$^{-2}$) solar panels, respectively (Table 4).

**Table 4.** Energy generation using Type A and Type B kinetic façade systems with optimal tilt angle ($\beta o$).

| Won-Hyo Elementary School in Seoul | | | | | | | | | | | | | |
|---|---|---|---|---|---|---|---|---|---|---|---|---|---|
| Month | | January | February | March | April | May | June | July | August | September | October | November | December | Total (kWh·Year$^{-1}$) |
| $\beta o$ (deg) | | 57° | 49° | 38° | 27° | 18° | 13° | 16° | 24° | 35° | 48° | 55° | 59° | |
| Type A (2 m × 1 m) | Total surface area 420 m$^{-2}$ | 482,289 | 492,432 | 490,076 | 472,775 | 446,618 | 428,713 | 439,793 | 465,352 | 487,650 | 493,277 | 485,885 | 479,129 | 5,663,989 |
| Type B (1 m × 2 m) | Total surface area 800 m$^{-2}$ | 888,354 | 894,496 | 881,317 | 837,777 | 779,963 | 741,741 | 762,753 | 819,075 | 870,400 | 895,327 | 889,141 | 883,338 | 10,143,682 |
| Cho-Rang Elementary school in Busan | | | | | | | | | | | | | |
| Month | | January | February | March | April | May | June | July | August | September | October | November | December | Total (kWh·Year$^{-1}$) |
| $\beta o$ (deg) | | 55° | 47° | 35° | 24° | 16° | 11° | 14° | 22° | 34° | 45° | 53° | 57° | |
| Type A (2 m × 1 m) | Total surface area 820 m$^{-2}$ | 1,026,250 | 1,043,950 | 1,035,148 | 986,746 | 933,387 | 891,238 | 917,938 | 975,667 | 1,030,998 | 1,043,969 | 1,031,271 | 1,031,271 | 11,947,833 |
| Type B (1 m × 2 m) | Total surface area 1640 m$^{-2}$ | 1,963,564 | 1,943,036 | 1,878,020 | 1,780,220 | 1,663,520 | 1,577,378 | 1,627,342 | 1,742,633 | 1,865,792 | 1,934,773 | 1,964,245 | 1,960,333 | 21,900,856 |

For the case of Won-Hyo Elementary School, the energy generation per unit area ($m^{-2}$) was 13,485 and 12,679 kWh·year$^{-1}$ for Types A and B solar panels, respectively. Whereas, for the case of Cho-Rang Elementary School, it was 14,570 kWh·year$^{-1}$ and 13,354 kWh·year$^{-1}$, respectively.

In both elementary schools, Type A produced more energy than Type B.

Figure 9 shows an image of the kinetic façade system installed on the Won-Hyo Elementary School and the installation process.

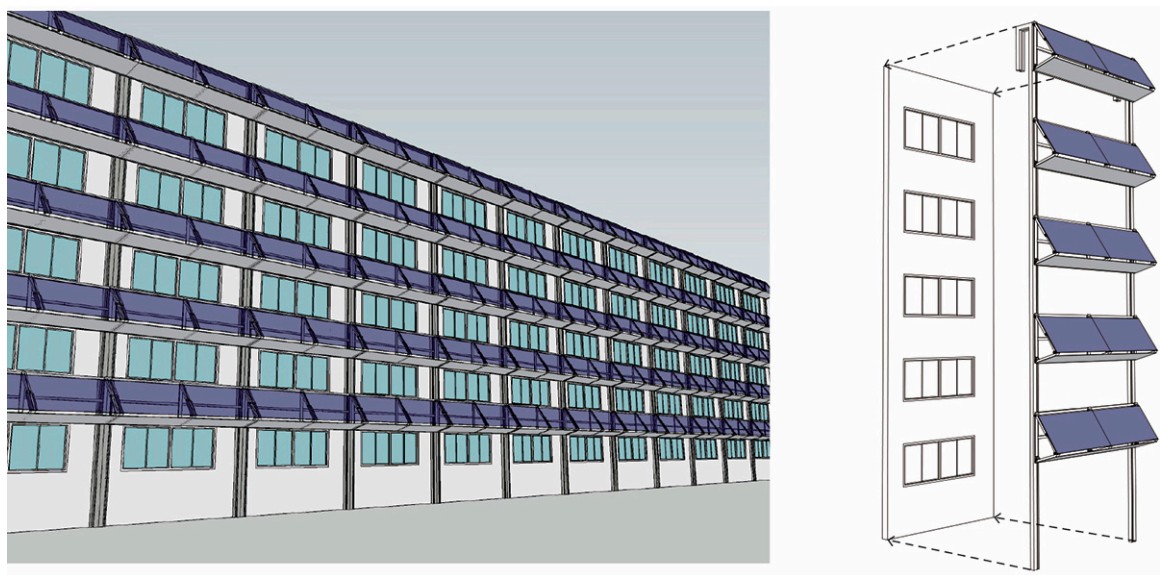

**Figure 9.** Simulation image of kinetic façade system installed on Won-Hyo Elementary School.

## 4. Results and Discussion

An architectural experimental façade design with solar panels for energy harvesting was applied to elementary schools in Seoul and Busan, South Korea. The design included a solar panel that could be adjusted each month according to the solar declination to maximize energy generation efficiency. According to an energy simulation based on the building elevation module, the best solar panel size was determined to be 2 m in width and 1 m in length. This façade design equipped with solar panel was installed on the elevation of the elementary schools facing south and southeast to receive optimal solar radiation.

In the case of Won-Hyo Elementary School in Seoul, with a total area of 420 $m^{-2}$, the façade system equipped with a solar panel was installed on the south- and southeast-facing building façades. The total energy production in 2021 was 5,663,989 kWh; the energy production per unit area ($m^{-2}$) of the solar panel was 13,485 kWh.

In the case of Cho-Rang Elementary School in Busan, with a total area of 820 $m^{-2}$, the façade system equipped with a solar panel was installed on the south- and southeast-facing building. The total energy production in 2021 was 11,947,833 kWh; the energy production per unit area ($m^{-2}$) of the solar panel was 14,570 kWh.

## 5. Conclusions

The proposed experimental façade design with monthly movable solar panels can maximize energy generation efficiency and can be installed in existing educational facilities as it has the same modular building elevation.

A new architectural design presented as a "kinetic façade", wherein the solar panel can be moved each month, is meaningful. A modularized solar energy harvesting kinetic façade system can be applied to existing and newly constructed educational facilities to increase energy independence and to large-scale building elevations, such as high-rise buildings in the future. A kinetic façade system can play an important role in creating a new architectural landscape.

The research on the driving device of the kinetic façade to make the solar panel move monthly has not been studied. A follow-up study is planned to address the limitations of this research. A follow-up study employing electrical energy and computer engineering researchers for economic analysis of renewable energy generation and development of a device that allows autonomous adjustment of the kinetic façade system according to the optical tilt angle has been planned.

**Funding:** This research was supported by the Gachon University Research Fund of 2020 (GCU-202008470009).

**Institutional Review Board Statement:** Not applicable.

**Informed Consent Statement:** Not applicable.

**Data Availability Statement:** Not applicable.

**Conflicts of Interest:** The author declares no potential conflict of interest with respect to the research, authorship and/or publication of this article.

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
