# Peer review of "Architectural Experiment Design of Solar Energy Harvesting: A Kinetic Façade System for Educational Facilities"

_applsci, doi:10.3390/app12125853_

Round 1

Reviewer 1 Report

Dear Author,

thank you for your work. The topic is really interesting - solar facades are one of crucial technologies from the standpoint of increasing energy efficiency of buildings. However, the quality of the manuscript is very poor and it cannot be published in the current form in the scientific journal. 

First of all, the organization of your paper is inappropriate. Manuscript should be divided into the following sections: Introduction (with literature review), Materials and Methods, Results, Discussion, Conclusions. 

The paper does not look as a scientific paper. You used Autodesk Revit 2020 software to calculate the parameters of the solar panels placed on the building wall. The scientific level is really poor. 

What about validation of your calculations?

What is new in your investigations compared to other available papers?

Reviewer 2 Report

The authors proposed an architectural design for renewable energy production to increase energy independence in the architectural field. To maximize renewable energy generation, solar panels can be adjusted according to the optimal tilt for each month. The solar panel developed in this study increases energy independence and presents a creative “kinetic façade,” in which solar panels move each month according to the optimal tilt angle. The manuscript would be suitable for publication after addressing the following comments/suggestions.

  1. In the abstract, the authors proposed a creative “kinetic façade”. However, the novelty of the kinetic façade is not clear, the reviewer suggested adding the relevant descriptions.
  2. Energy harvesting is a hot issue, not only solar energy, but also wind energy, mechanical energy, etc. It is suggested that the author improve the first two paragraphs of the introduction. These are references: Advances in Applied Energy, 2022, 6: 100091. Applied Energy 239, 735-746. Mechanical Systems and Signal Processing 169, 108637
  3. There are too few experimental data. The author suggests adding experimental data of different cases.
  4. There are some drawing errors and writing mistakes that can be avoided. Also, the language of the article is not logical enough and needs to be further written and improved.

Reviewer 3 Report

1) I hope author should aware that there are two Korea's in the world. Therfore,  requested to add Sout Korea wherever, applicable

2) Table 1 citation number 5 as a reader I cross verified but the link is not available requested to cross check and made available readers

3) Figure 1 requested to explain more in detail as a reader I did not see any scientific justification for figure 1

4) Section 3.1 very well explained but I feel more scientific citations needed for that I suggest few more open access articles to cite 

a. Effect of dual surface cooling of solar photovoltaic panel on the efficiency of the module: experimental investigation

b. Experimental Investigation of the Effect of a Combination of Active and Passive Cooling Mechanism on the Thermal Characteristics and Efficiency of Solar PV Module

5) I hope author should know difference between Article and Communication. The article " Architectural Experiment Design of Solar Energy Harvesting: A Kinetic Façade System for Educational Facilities " submitted to the as a reader I feel it is communication because there is a no discussions of results & discussion section. In addition,  it's just an informative article or case study to the ministry of education 

6) As a reader, in line 191-192 I feel to ask a question that author stated that  "The design included a solar panel that can be adjusted each month according to the solar declination to maximize energy generation efficiency". Therefore, I will ask my question point to point in tablaur form 

6a) The design included a solar that can be adjusted each month "How it will adjust"? and "who will adjust"?

6b) If the adjustment is according to the solar tracking mechanism requested to do analysis 

6c) if it is manually then what is the addition of bucks need to be spend is not stated well

7) what is the total area of Won-HYO school not stated?

8) Moreover, author stated in line 55-56 South Korean's building are built according to the standard blueprints issued by ministry of education but my question is  what about the area between school to school, is it also same?.   

Round 2

Reviewer 3 Report

Dear Authors

Thanks for sending your response comments. 

In line 239-241 “solar panel that could be adjusted each month according to the solar declination to maximize energy generation efficiency” if it is limitation of the study then add a new section to the conclusions part what are the limitations of the current study (Minimum 4-5 lines with proper justifications)? Therefore, other researcher’s and Ministry of Education, South-Korea government will understand that the authors have been limited the present research work. Otherwise, authors are requested to respond my questions, as presented below

1a) The design included a solar that can be adjusted each month "How it will adjust"? and "who will adjust"?

1b) If the adjustment is according to the solar tracking mechanism requested to do analysis 

1c) if it is manually then what is the addition of bucks need to be spend is not stated well 
